# Galectin-3 as a Next-Generation Biomarker for Detecting Early Stage of Various Diseases

**DOI:** 10.3390/biom10030389

**Published:** 2020-03-03

**Authors:** Akira Hara, Masayuki Niwa, Kei Noguchi, Tomohiro Kanayama, Ayumi Niwa, Mikiko Matsuo, Yuichiro Hatano, Hiroyuki Tomita

**Affiliations:** 1Department of Tumor Pathology, Gifu University Graduate School of Medicine, Gifu 501-1194, Japan; 2Medical Science Division, United Graduate School of Drug Discovery and Medical Information Sciences, Gifu University, Gifu 501-1193, Japan

**Keywords:** galectin-3, biomarker, diagnostic, prognostic, early stage, tumor, animal model

## Abstract

Galectin-3 is a β-galactoside-binding lectin which is important in numerous biological activities in various organs, including cell proliferation, apoptotic regulation, inflammation, fibrosis, and host defense. Galectin-3 is predominantly located in the cytoplasm and expressed on the cell surface, and then often secreted into biological fluids, like serum and urine. It is also released from injured cells and inflammatory cells under various pathological conditions. Many studies have revealed that galectin-3 plays an important role as a diagnostic or prognostic biomarker for certain types of heart disease, kidney disease, viral infection, autoimmune disease, neurodegenerative disorders, and tumor formation. In particular, it has been recognized that galectin-3 is extremely useful for detecting many of these diseases in their early stages. The purpose of this article is to review and summarize the recent literature focusing on the biomarker characteristics and long-term outcome predictions of galectin-3, in not only patients with various types of diseases, but associated animal models.

## 1. Introduction

Galectins are a family of widely expressed β-galactoside-binding lectins in modulating “cell-to-cell” and “cell-to-matrix” interactions in all organisms [1,2,3]. Mammalian galectins have either one or two highly conserved carbohydrate recognition domains (CRDs), recognizing β-galactoside residues, to form complexes that crosslink glycosylated ligands [4,5,6].

Galectins have been classified into three subgroups according to their CRD number and function (Figure 1): (1) Proto-type galectins (galectin-1, -2, -5, -7, -10, -11, -13, -14, and -15), containing a single CRD that form non-covalent homodimers, (2) tandem-repeat galectins (galectin-4, -6, -8, -9, and -12), carrying two CRD motifs connected by a peptide linker, and (3) chimera-type galectin (galectin-3), which are characterized by having a single CRD and an amino-terminal polypeptide tail region [2,5,6]. All members of galectin family were numbered consecutively by order of discovery (Figure 1). Galectins are ubiquitously present in vertebrates, invertebrates, and also protists [1].

Galectins play important roles in cell-to-cell and cell-to-matrix interactions by binding to endogenous glycans. Galectin signaling can regulate cellular functions at the cell surface. Biological functions of galectins, which are not yet fully understood, include roles in development, tissue regeneration, regulation of immune cell activities, and other important cellular functions [1,2].

Galectins are important regulators of inflammatory responses and immune system. In fact, galectins are expressed in many inflammatory cells, such as macrophages [7]. Depending on the inflammatory environment, galectins promote pro-inflammatory or anti-inflammatory responses [2,5,6]. Recently, the galectin-mediated specific molecular recognition of glycans on the cell surface have revealed their roles as innate immune functions against potential pathogens and parasites. As a part of the innate immune system for microbial recognition/effector functions, galectins bind to exogenously exposed glycans on the surface of viruses, bacteria, fungi, and parasites [3].

## 2. Clinical Significance and Applications of Galectine-3 (Gal-3)

Gal-3, approximately 30 kDa chimera-type galectin, is expressed in the nucleus, cytoplasm, mitochondrion, cell surface, and extracellular space. Gal-3, to an equal or greater extent than other galectins, plays an important role in cell-to-cell or cell-to-matrix interactions, cell growth and differentiation, macrophage activation, antimicrobial activity, angiogenesis, and apoptosis [8]. Extracellular and intracellular activities of Gal-3 are shown in Figure 2, with references detailed in the caption.

Recently, Gal-3 has been indicated to be involved in the following broad pathological processes: Inflammation [9], fibrosis, cell-to-cell [10,11] or cell-to-matrix [12] contacts, cell proliferation [13,14], and protection from apoptosis [15,16]. Furthermore, many studies have revealed that Gal-3 expression is detected in many disease conditions, such as heart disease [17,18,19,20,21,22,23], kidney disease [24,25,26,27], diabetes mellitus [24,25,28], viral infection [29,30,31,32], autoimmune disease [33,34,35,36], neurodegenerative disorders [37,38,39,40,41,42], and tumor formation [43,44,45,46,47,48,49,50,51,52].

Since Gal-3 is readily secreted to the cell surface and into biological fluids (e.g., serum and urine), from injured cells and inflammatory cells, Gal-3 may be used as a sensitive diagnostic or prognostic biomarker for various pathological conditions [44,52,53,54]. Furthermore, Gal-3 may also be useful for detecting very early stage of some diseases. Gal-3 has already been used as a novel biomarker in the early detection of myocardial dysfunction and heart failure [55]. Gal-3 has been validated as a biomarker of fibrotic degeneration in acute myocarditis following cardiac viral infection. In an animal model of heart failure, serum Gal-3 levels was shown to be used as an diagnostic biomarker for early detection of cardiac degeneration in acute myocarditis [31] and acute myocardial infarction [56]. Furthermore, Gal-3 is also a good specific marker for indicating the early stage of glioma tumorigenesis [43].

Serum levels of Gal-3 are usually determined by enzyme-linked immunosorbent assay (ELISA) using commercially available materials [57,58]. The localization or time-course of Gal-3 expression in various organs are examined immunohistochemically using commercially available anti-Gal-3 antibodies [31]. Gal-3 levels may be changed by different clinical factors in patients, depending on the underlying pathological conditions. Some previous studies indicated contradicting results on the association between Gal-3 levels and the prediction of clinical outcomes [59,60,61]. These studies, however, might be limited by the insufficient size of the clinical sample. In this review, we discuss and summarize recent developments of the biomarker characteristics and long-term outcome prediction of Gal-3 in patients with various types of diseases (Table 1) as well as associated animal models (Table 2). Furthermore, we provide an overview of Gal-3 as a next-generation biomarker for detecting early stage of various diseases.

## 3. Gal-3 and Heart Disease

### 3.1. Clinical Use of Gal-3 as a Possible Biomarker in Heart Disease

Recently, many reports have indicated that Gal-3 is a novel biomarker of heart disease and a prognostic indicator of patients with heart failure.

Clinical studies suggest that serum and myocardial Gal-3 levels in patients with heart failure reflects cardiac inflammatory condition and can be considered as a useful biomarker for both cardiac inflammation and fibrosis, depending on the pathophysiological mechanisms of heart failure [93]. In the general population, a high concentration of plasma Gal-3 correlates with clinical outcomes of heart failure [94,95]. The increased serum levels of Gal-3 are associated with adverse clinical events in both patients with acute [62,63] and chronic [64,65] heart failure with preserved or reduced ejection fraction.

### 3.2. Established Biomarkers in Heart Disease

Some practical guidelines have indicated the utility of established or recommended biomarkers in diagnosis and risk management of heart failure. In fact, American College of Cardiology Foundation (ACC)/American Heart Association (AHA), Heart Failure Society of America (HFSA), and European Society of Cardiology (ESC) have established the natriuretic peptides (NPs), circulating hormones of cardiac origin that play an important role in the regulation of intravascular blood volume and vascular tone, as useful diagnostic biomarkers in the patients suspected of heart failure [96,97,98,99]. However, the utilities of novel biomarkers other than NPs are less established in clinical practice. The recommendations on the clinical assessment and analytical perspectives of novel biomarkers in diagnosis and management of heart failure were prepared by the National Academy of Clinical Biochemistry (NACB) [100]. In these criteria, novel biomarkers need to be able to recognize fundamental causes of heart failure, assess its severity, and foresee the risk of the disease progression. The ACC/AHA guidelines recommended the use of Gal-3 as a biomarker for assessment of myocardial fibrosis in heart failure, however, the ESC has not recommended the clinical use of Gal-3 [101].

### 3.3. Gal-3 as a Biomarker of Fibrosis

Myocardial fibrosis is tightly implicated in the pathophysiological mechanisms for ventricular remodeling of heart failure. As a potential key factor in pathophysiology of heart failure, US Food and Drug Administration has approved Gal-3 as soluble biomarkers for myocardial fibrosis to detect ventricular remodeling [96]. Thus, the serum levels of Gal-3 are associated with ventricular remodeling and cardiac function. However, whether and how Gal-3 plays a pathophysiological role in cardiac remodeling remains unclear, especially in clinical settings. In fact, the serum levels of Gal-3 was not related to left ventricular remodeling defined by cardiac MRI in patients with acute myocardial infarction and left ventricular dysfunction, although certain biomarkers involved in extracellular matrix turnover, such as matrix metalloproteinase-3 and monocyte chemoattractant protein-1 at baseline (mean 46 h), were highly associated [66]. In addition, Srivatsan et al. conducted a study that found that Gal-3 could not predict mortality, while other biomarkers could be used as a predictor of mortality [102].

### 3.4. Usefulness of Gal-3 in Animal Models

As mentioned above, the roles of Gal-3 in heart failure and cardiovascular disease are still controversial clinically, however, many animal models clearly demonstrate the possibility of Gal-3 as a novel biomarker of heart disease. Overexpression of myocardial Gal-3 during early pre-symptomatic stages of heart failure has been well documented in several studies using animal models. Intrapericardial injection of recombinant Gal-3 in healthy rats significantly increased the degree of myocardial fibrosis with ventricular remodeling, and the induction of heart failure [80,81]. And Gal-3 was indicated to colocalize with activated myocardial macrophages [80]. On the other hand, cardiac remodeling and dysfunction induced by Gal-3 was prevented by a pharmacological inhibitor of Gal-3, N-acetyl-seryl-aspartyl-lysyl-proline [80]. In a rat model of heart failure, increased Gal-3 was detected at hypertrophied hearts, prior to the development of heart failure [81]. These findings imply that Gal-3, at the early stages of inflammatory responses, may be a potential therapeutic target in the treatment of heart failure. A rat model of right ventricular heart failure, created by banding of the pulmonary artery, demonstrated that Gal-3 was significantly increased in both right and left ventricles and that protein kinase C promoted cardiac fibrosis and heart failure by modulating Gal-3 expression [82]. In order to clarify the important role of myocardial Gal-3 expression during the early stage of heart failure, the time-course analysis of cardiac and serum Gal-3 in viral myocarditis, which was induced at 12, 24, 48, 96 h, 7 and 10 days after specific virus-inoculation, was performed using an animal model [31]. Gal-3 was indicated as a useful histological biomarker of cardiac fibrosis in acute myocarditis following viral infection and serum Gal-3 levels might be an early diagnostic method for detecting cardiac fibrotic degeneration in acute myocarditis [31].

However, the clinical data has not shown that circulating Gal-3 levels reflect myocardial Gal-3 levels or myocardial fibrosis, although circulating Gal-3 has been demonstrated as a potential predictor for clinical outcome in several cohort studies, as mentioned earlier [94,95]. In clinical settings, since various stages and degrees of myocardial inflammation and fibrosis may be present in a patient with heart disease, serum Gal-3 levels may reflect a sum of different pathophysiological conditions [93]. Thus, it may be that the circulating Gal-3 levels in a patient with various stages of heart disease will not adequately reflect myocardial inflammation and fibrosis.

### 3.5. Clinical Use of Gal-3 as a Next-Generation Biomarker in the Future

Endomyocardial biopsy is widely used as a diagnostic tool for patients with heart diseases, such as myocarditis and other cardiomyopathies, which are often difficult to diagnose by conventional radiological imaging methods [103]. The histologic examination of endomyocardial biopsy is still the gold standard for final diagnosis, such as for myocarditis and other cardiomyopathies like cardiac fibrosis, despite the continuous advancements in diagnostic and therapeutic approaches [104,105,106].

Unlike uniform materials from experimental animals, there are many variables in human biopsy materials by its nature. Human biopsies are usually obtained under different conditions, e.g., variable time periods between biopsy and processing, as well as variation in disease onset or severity. In contrast, the serum levels of Gal-3 reflect myocardial Gal-3 expression or cardiac fibrosis by using a sophisticated animal model for time-course histological examination [31]. It is possible that the difference between experimental data and clinical findings is due to a wide variability in clinical settings with differences in sample collection and disease stages. Since experimental animal data clearly indicate that serum Gal-3 might be an early diagnostic biomarker for cardiac degeneration or fibrosis in acute myocarditis [31], further studies are thus needed to investigate whether such findings are also applicable to cardiac degeneration or fibrosis in humans. Gal-3 can be used as a predictive biomarker for the early stage or new-onset of heart failure, especially if it is only the first single pathological factor. In addition, Gal-3 can also be used as an additional indicator for detecting worse prognosis, mortality, and readmission.

## 4. Gal-3 in Various Nervous System Processes

### 4.1. Gal-3 in Microglial and Macrophages

Gal-3 is associated with microglial activation and proliferation in various neuropathological lesions, such as traumatic brain injury [107,108], ischemic insult [37,39], demyelination [84], and encephalitis [32]. Gal-3 is a unique marker of activated resident microglial and recruited macrophages from blood circulation [42]. Both of the mesodermal origin immune cells have capacity for myelin phagocytosis. The co-localization between Gal-3 and myelin phagocytosis strongly suggests that Gal-3 indicates a unique state of activation in microglia and macrophages that correlates with their ability to phagocytose myelin [42]. In fact, Gal-3 is always expressed in both activated microglia and recruited macrophages that phagocytose degenerated myelin, but not in non-activated microglia that do not phagocytose myelin. However, whether Gal-3 induction is also instrumental in activating phagocytosis is still unknown.

### 4.2. Gal-3 as a Possible Biomarker in Neurodegenerative Disease

Oligodendrocytes in the central nervous system make the myelin sheath surrounding neuronal axons and provide trophic support and electrical insulation for axonal conduction. Myelin degeneration is the damage of the myelin sheath surrounding axons and the impairment of the conduction of signals in the affected nerves. Myelin degeneration can occur in various neuropathological processes, such as traumatic brain injury, ischemic brain damage, and encephalitis. These pathogenic areas coincide with the lesions corresponding to microglial activation. In fact, microglial activation is also observed in these demyelinating lesions, and thus Gal-3 expression is also associated with such lesions because the removal of degenerated myelin by activated microglia and macrophages is essential for regeneration of myelin sheath after axonal injury [109,110].

Experimental autoimmune encephalomyelitis (EAE) is a widely used animal model of myelin degeneration in CNS, and is regarded as a model of autoimmune-mediated demyelinating multiple sclerosis in humans. Gal-3-deficiency in mice was reported to be responsible for the reduction in severity of EAE [83]. Gal-3 was clearly observed in injured white matter of spinal cord in EAE mouse model at early stage on day 12 after immunization with complete Freund’s adjuvant and MOG peptide, which is before paralysis-starting stage in the fore- and hind-limbs between days 18 and 21 [84]. These results suggest that Gal-3 plays an important pathogenic role in multiple sclerosis, and it may be a potential therapeutic target for autoimmune demyelinating diseases.

Recently, it was reported that Gal-3 is also associated with other neurodegenerative diseases. In a mouse model of Huntington’s disease, the Gal-3 expression was already up-regulated before motor impairment, and the expression level remained high in activated microglia throughout disease progression [40]. It is suggested that suppression of Gal-3 improves microglia-mediated neurodegenerative lesions, and that Gal-3 is a novel therapeutic target for Huntington’s disease. Parkinson’s disease is another neurodegenerative disease, which is caused by the loss of specific neurons secreting the neurotransmitter dopamine in the basal ganglia. It is reported that serum Gal-3 may be a potential marker for the identification of Parkinson’s disease and advanced clinical stages [111].

### 4.3. Gal-3 Is Associated with Cerebrovascular Disease

Some clinical studies indicated that higher serum levels of Gal-3 correlate with worse outcomes in stroke and cerebrovascular diseases. The studies raise the possibility that Gal-3 is closely linked to the inflammatory cascade within the central nervous system following cerebrovascular attack and can be a prognostic marker and therapeutic target in cerebrovascular diseases [112,113].

Yan et al. [67] found that a group of 110 Chinese patients with acute intracerebral hemorrhage had significantly higher Gal-3 levels in their plasma than a group of 110 controls. It was an independent prognostic predictor for 1-week mortality, 6-month mortality, and 6-month unfavorable outcome. Elevated plasma Gal-3 levels are strongly associated with the inflammation and severity after intracerebral hemorrhage, and Gal-3 was identified as a prognostic biomarker for intracerebral hemorrhage. Liu et al. [68] evaluated plasma Gal-3 levels in 120 Chinese patients with subarachnoid hemorrhage caused by ruptured aneurysm, and compared them with 120 healthy controls. Similar to intracerebral hemorrhage, patients showed significantly higher Gal-3 levels as compared to controls. Their findings indicated there was an association between Gal-3 levels and the severity of subarachnoid hemorrhage and worse prognosis. Thus, their results suggested that Gal-3 levels could be an independent prognostic marker after subarachnoid hemorrhage.

Few studies have examined the relationship of Gal-3 with global brain ischemia. Sävman et al. [69] reported that Gal-3 was increased in cerebrospinal fluid collected from human infants who had undergone birth asphyxia from various causes and had demonstrated severe clinical course and adverse outcomes.

In animal studies, time course analysis of Gal-3 expression in hippocampal CA1 sector of gerbils following transient forebrain ischemia revealed increase of Gal-3, which was maximal at 96 h after reperfusion [39]. The time of 96 h after reperfusion is very important in this situation, because the peak of the neuronal apoptosis in hippocampal CA1 occurs around the same time [114]. The expression of Gal-3 was strongly reduced by hypothermia during ischemic insult and thus the hypothermic prevention of Gal-3 inhibited microglial activation [39]. The results indicate an inhibitory effect of the neuronal damage by the modulating of Gal-3 expression by hypothermia.

A temperature-dependent enhancement of Gal-3 expression in microglia in addition to the acceleration of apoptotic neuronal DNA fragmentation, which was observed in mild hyperthermia between 37 °C and 39 °C, occurred in hippocampal CA1 region following transient forebrain ischemia [37,39]. In these study cases, Gal-3 expression was localized within CA1 region and observed only in cells which expressed Iba-1, meaning that microglia in CA1 region predominately contributed to the Gal-3 expression. In contrast to these animal models of transient ischemia contributing to the specific hippocampal neuronal death, in another animal model (permanent cerebral infarction using middle cerebral artery occlusion in rats), there was persistent up-regulated expression of Gal-3 in the ischemic lesions at day 1 after occlusion; the number of Gal-3 positive cells in the cerebral infarction was further increased on days 2 and 3, and peaked at day 7 after occlusion [85]. The up-regulated expression of Gal-3 was detectable even 2 months after middle cerebral artery occlusion.

The differences of the peak time of Gal-3 expression in animal models suggest that different mechanisms exist at different times or stages of ischemic neuronal damages. Hippocampal neuronal death in the CA1 region following transient forebrain ischemia showed acute activation only in resident microglia without recruited macrophages [37,39]. The permanent cerebral infarction by middle cerebral artery occlusion, however, demonstrated up-regulated expression of Gal-3 in several cell types, including activated microglia, infiltrating macrophages and activated astrocytes [85]. The late peak time and persistent Gal-3 expression in the latter animal model reflects the sum of Gal-3 expression in activated microglia, infiltrating macrophages and activated astrocytes. Contrary to the more complex expression of Gal-3 in the late stage of permanent ischemia, the usefulness of the early stage of Gal-3 analysis for ischemic brain damages, which consist of mainly activated microglia, is clearly shown.

## 5. Gal-3 in Renal Disease

### 5.1. Cardiovascular Disease and Chronic Kidney Disease

Many cardiac biomarkers which reflect cardiac inflammation and fibrosis may also contribute to the progression of kidney disease. It is plausible that cardiovascular disease (CVD) and chronic kidney disease (CKD) are closely interrelated with each other, and patients with CKD have a strong risk of CVD [115,116]. It is well known that CKD is prevalent in patients with CVD and responsible for approximately half of all CKD-related deaths [26]. Thus, patients with end-stage CKD are at much higher risk of mortality due to CVD. As mentioned earlier in Section 3, “Gal-3 and heart disease”, Gal-3 is detected in cases of myofibroblast proliferation, fibrogenesis, tissue repair and myocardial remodeling, and is also associated with kidney fibrosis and increased risk of CKD. The wide tissue distribution of Gal-3 expression associated with fibrosis in both heart disease and kidney disease complicates the utility of Gal-3 as a cardiac biomarker in CKD patients [26].

### 5.2. Gal-3 as a Biomarker of CKD in Clinical Studies

Higher concentrations of Gal-3 may be associated with progression of CKD, indicating potential novel mechanisms related to Gal-3 expression that may contribute to the progression of CKD [25]. Furthermore, Gal-3 is reported to play a pivotal role in renal interstitial fibrosis and progression of CKD [117]. Thus, inhibition of Gal-3 may be a promising therapeutic strategy to prevent advanced renal disease. Kang et al. [70] examined Gal-3 expression in renal biopsy specimens of 88 patients with systemic lupus erythematosus (SLE) nephritis and in five normal controls, and serum Gal-3 levels were also measured in 20 patients with SLE, including 11 with nephritis. Glomerular Gal-3 expression was observed in 81.8% of patients with SLE nephritis but not in five controls. Serum Gal-3 levels were particularly higher in SLE patients with nephritis than in healthy controls. Thus, Gal-3 may contribute to the glomerulonephritis in SLE.

### 5.3. Gal-3 in Animal Data for CKD

Studies in animal models demonstrate that Gal-3 plays a pivotal role in several nephropathies [36]. In several animal models of CKD, inhibition or suppression of Gal-3 has a protective effect on CKD outcomes. However, some animal models show even opposite effects depending on the stage and degree of tissue inflammation and injury. Gal-3 deficient mice presented less acute renal tubular necrosis and a more prominent tubular regeneration in renal tissue damage triggered by ischemia and reperfusion injury, when compared with controls [86]. Absence of Gal-3 protected against activated myofibroblast accumulation and fibrosis in experimental hydronephrosis mouse model induced by unilateral ureteric obstruction [87]. However, the opposite results were also reported in other studies [88,89]. In a mouse model for chronic kidney disease induced by unilateral ureteral obstruction, Gal-3 not only protected renal tubules from chronic injury by limiting apoptosis, but also enhanced matrix remodeling and fibrosis attenuation [88]. Fibrotic and inflammatory changes in lipid-induced glomerulosclerotic injury, which was induced by feeding an atherogenic high-fat diet, were significantly more marked in Gal-3 deficient mice [89].

## 6. Gal-3 in Other Diseases

Expression of Gal-3 may be associated with many diseases other than heart disease, neurodegenerative disease, cerebrovascular disease, and renal disease described above. Numerous studies have indicated that Gal-3 may also be used as a diagnostic or prognostic biomarker for these disease conditions [53,118].

### 6.1. Assessing Liver Fibrosis

Assessing liver fibrosis, which is usually performed by liver biopsy as a gold standard method, is important for determining disease management and additional treatment strategy. Gal-3 is expressed in Kupffer cells during progression of liver fibrosis, and recently Gal-3 related binding protein is reported as a serum surrogate marker for assessing liver fibrosis [71]. Experimental Gal-3 deficient mice of dietary-induced nonalcoholic fatty liver disease (NAFLD)/nonalcoholic steatohepatitis (NASH) were used for elucidating the pathogenesis of NASH and liver fibrosis [90]. The Gal-3 deficient mice, when fed an obesogenic high-fat diet, developed adiposity, hyperglycemia, and hepatic steatosis, whereas the liver tissues showed attenuated inflammation and fibrosis. Furthermore, a possible involvement of Gal-3 with Kupffer cell activation during Th1/Th2 immune responses, granulofibrous reaction, and tissue repair is strongly indicated in schistosomiasis-induced liver fibrosis [119].

### 6.2. Rheumatoid Arthritis and Osteoarthritis

Gal-3 has been reported to be expressed and secreted by inflamed synovium in patients of rheumatoid arthritis and osteoarthritis. Issa and colleagues reported that Gal-3 serum levels was persistently increased in early rheumatoid arthritis, and a positive correlation was observed between Gal-3 serum levels and some associated pathophysiology, such as autoimmunity, smoking, and joint destruction [72,120]. In experimental animal models for osteoarthritis, Gal-3 has been demonstrated to induce joint swelling and osteoarthritis-like lesions after intra-articular injection [119].

### 6.3. Connective Tissue Diseases

In patients with systemic lupus erythematosus (SLE), the immune response mediated by serum anti-Gal-3 antibody plays a key role in the pathogenesis of SLE skin lesions [73]. The pathogenicity of serum anti-Gal-3 antibody was investigated using an animal experimental model. Administration of purified serum anti-Gal-3 antibody to female BALB/c mice induced lupus-like histologic changes [73].

The serum concentration of Gal-3 is associated with fibrosis and inflammation in systemic sclerosis, which is characterized by progressive fibrosis of the skin and certain internal organs, and may be a prominent biomarker of disease activity [121]. Recently, Faludi et al. [74] suggested that Gal-3 is an independent predictor of all-cause mortality as well as cardiovascular mortality in systemic sclerosis patients, and that the serum levels of Gal-3 are associated with advanced organ fibrosis and inflammation.

### 6.4. Skin Diseases and Its Repair

Gal-3 is expressed on keratinocytes, hair follicles, sebaceous, and sweat glands in the skin. Furthermore, it is also expressed by other migrating cells into the skin, including dendritic cells, fibroblasts, and monocytes [122]. Deletion of Gal-3 in mouse models shows different inflammatory or remodeling effects on skin compared to other organs such as heart, lung and kidney [123]. For example, in psoriasis patients, epidermal Gal-3 expression is significantly decreased in the psoriatic skin lesions, but not in non-psoriatic skin lesions. However, these findings are not observed in related diseases known as psoriasiform keratosis clinically and histologically similar to psoriasis [124].

Advanced glycation end products (AGEs), known as harmful compounds, seem to contribute to defective skin repair in not only diabetic patients but a result of normal aging process. An inverse correlation between AGEs and Gal-3 localization has been reported in wound healing. And thus, it is suggested that Gal-3 may protect against accumulation of AGEs in the wound healing [125].

These findings indicate that Gal-3 may be a novel therapeutic target for a variety of skin diseases, however, there are few reports of Gal-3 as a convenient serum biomarker because of its ubiquitous localization of skin.

## 7. Gal-3 in Neoplasms

### 7.1. Gal-3 in Tumors and the Prognosis

Abnormal expression of tumor-associated galectins correlates with the development, progression and clinical aggressiveness of the tumors, as well as the contribution to metastatic phenotype [54,126,127,128,129,130,131]. Tumor-derived galectins have opposite functions on different immune cells that either promote inflammation or decrease immune responses, depending on the inflammatory environment among tumor cells. This means that the tumor-derived galectins have bipotential consequences on both tumor and immune cells [130]. Thus, new therapeutic strategies may be designed to facilitate the use of galectins as biological response modifiers to either tumor cells or immune cells [54,129,131].

There are many reports that indicate the association between the expression of Gal-3 in tumors and the prognostic value in patients. The prognostic value of Gal-3, however, seems to be tumor-type-dependent. In general, expression of Gal-3 is up-regulated in many types of cancers, however, there are variable results in studies reporting either increased or decreased survival rates [53]. In other galectins, for example, increased galectin-1 expression is reported to associate with poor prognosis, whereas elevated galectin-9 expression is considered as a marker of favorable prognosis [132]. Gal-3 may not be considered a tumor diagnostic biomarker, but rather a tumor function-related marker, which may be used in combination with other metabolic or immunological biomarkers. Abnormal expression of Gal-3 in neoplasms have been summarized in Table 1 for clinical cases and in Table 2 for animal models.

### 7.2. Gal-3 in Tumor-Associated Macrophages

Macrophages are the most abundant immune cell population present in tumor tissue, and these cells are called tumor-associated macrophages (TAMs) [133,134]. The density of TAMs is reported to be associated with a poor prognosis and positively correlated with tumor progression in several studies. Gal-3, like other cytokines and proteins, such as CCL-17, CCL-22, TGF-β, IL-10 and arginase 1, is expressed by TAMs [133]. Gal-3 is secreted into tumor stroma and plays an important role of tumor microenvironment contributing to the tumor progression. Gal-3 also affects the tumor angiogenesis by regulating vascular endothelial growth factor [135].

Macrophages are generally categorized as M1 or M2 types, and the majority of TAMs have been shown to exhibit M2-like phenotype in the tumor microenvironment [134]. As a rule, M1 macrophages are considered to have inflammatory and cytotoxic function, whereas M2 macrophages are anti-inflammatory and promoting wound healing. Both M1- and M2-macrophage phenotypes are Gal-3 positive and participate in liver cirrhosis through the production of both M1- and M2-related factors in a rat model [136]. In parasite-infected brains, using an animal model of neurocysticercosis, macrophages/microglia with the M1-activation phenotype are thought to promote brain damage, whereas macrophages/microglia with the M2-activation phenotype are thought to play important anti-inflammatory and tissue reparative functions [137]. Furthermore, the parasite-infected brain shows abundant Gal-3 expression in M2 macrophages, and the Gal-3 in M2 macrophages plays a central role in protective function by clearance of neutrophils accumulated in the brain. However, in tumor microenvironment, M2 macrophages which are major component of TAMs promote tumor progression and contribute to resistance to chemotherapies [133,134].

### 7.3. Gal-3 in Early Neoplastic Lesions

In the ethylnitrosourea-induced rat glioma model, Gal-3 was clearly expressed in glial early neoplastic proliferation (ENP) and glial microtumor but rarely in microglia [43]. In malignant gliomas, which developed from ENP and microtumors, Gal-3 was expressed in both neoplastic glioma cells and microglia/macrophages including TAMs [43]. This suggests that Gal-3 is activated in not only glial tumor development but microglia/macrophages through the progression of glioma. Thus, Gal-3 is a useful glioma-specific marker, only in the early stage of glioma tumorigenesis. Gal-3 has also been shown as a glioma related marker in human gliomas and expression of Gal-3 has been reported to correlate with WHO tumor grade [138]. In humans, Gal-3 is not a glioma-specific marker, since early neoplastic lesions such as ENP are still unknown and Gal-3 is expressed in both neoplastic glioma cells and microglia/macrophages including TAMs. Thus, Gal-3 detected in early neoplastic lesions of animal models cannot be applied currently as an early diagnostic marker for clinical use in either histopathological assessment or serum analysis.

### 7.4. Complicated Mechanism of Gal-3 Expression in Neoplasms

As mentioned above, Gal-3 expression in neoplasms, especially in malignant tumors, is complex. Gal-3 is produced by not only tumor cells but also tumor-related immune cells including TAMs. Furthermore, main cells producing Gal-3 are changed gradually from the early stage of tumor development to the advanced stage of tumor progression. Gal-3 is detected in not only localized tumor cells, but also constantly recruited immune cells from the circulation. Moreover, both M1 macrophages, showing mainly inflammatory and cytotoxic function, and M2 macrophages, possessing anti-inflammatory and tumor-promoting activity, are Gal-3 positive through the tumor progression.

## 8. Conclusions and Perspectives

Many studies have revealed that Gal-3 expression is detected in various disease conditions, such as heart disease, kidney disease, diabetes mellitus, viral infection, autoimmune disease, neurodegenerative disorders, and tumor formation. In addition, Gal-3 is detected in not only localized tumor cells but also constantly recruited immune cells from the circulation. Moreover, M1 macrophages, showing mainly inflammatory and cytotoxic function, and M2 macrophages, possessing anti-inflammatory and tumor-promoting activity, are both Gal-3 positive through the tumor progression. On the other hand, Gal-3 is easily secreted to the cell surface and into biological fluids like serum and urine, and it is also released from injured cells and inflammatory cells. Thus, Gal-3 may be used as a sensitive diagnostic or prognostic biomarker under various pathological conditions.

Since Gal-3 levels are altered by different clinical factors depending on the underlying pathological conditions in patients, Gal-3 itself is not a disease-specific marker, however, Gal-3 is a potential biomarker for targeting early stage of several diseases. Furthermore, Gal-3 at the early stages of inflammatory responses may be a potential therapeutic target for the diseases, especially in cardiac fibrosis, autoimmune diseases, neurodegenerative diseases, and cerebrovascular diseases. The new therapeutic strategies may also be designed to prove the clinical efficacy of Gal-3 as biological response modifiers to either tumor cells or tumor-related immune cells.

## Figures and Tables

**Figure 1 biomolecules-10-00389-f001:**
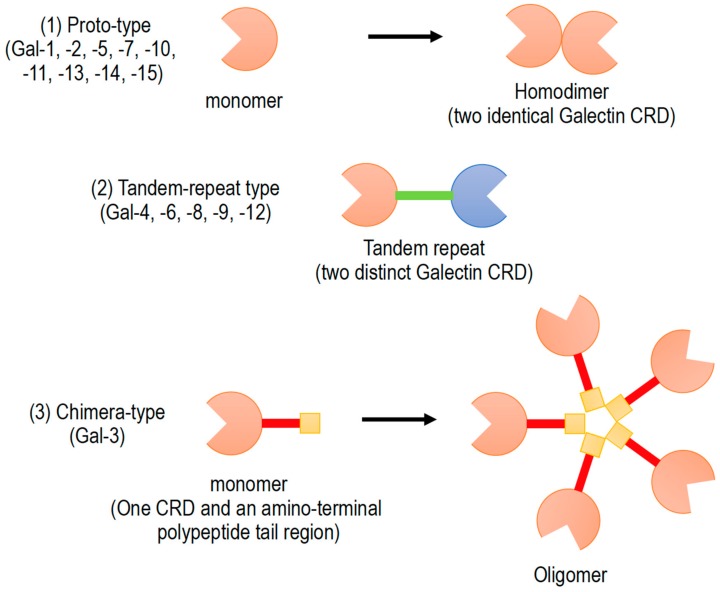
Schematic diagram of galectin family members. Galectin members are divided into three types based on the organization of galectin carbohydrate recognition domain (CRD).

**Figure 2 biomolecules-10-00389-f002:**
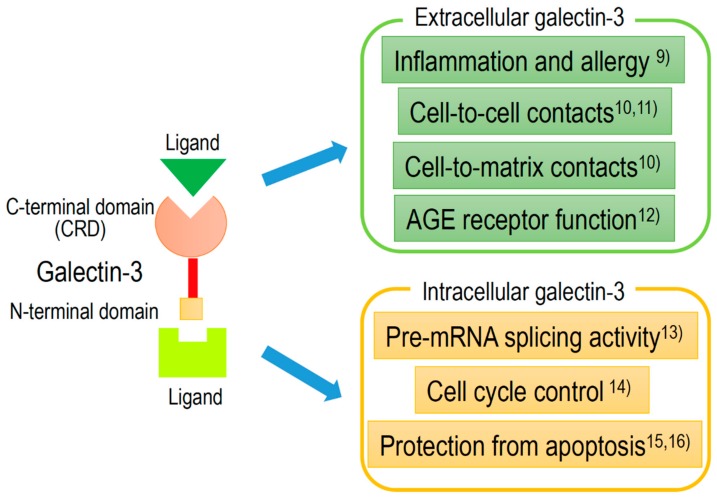
Schematic structure of galectin-3 and the intracellular and extracellular functions.

**Table 1 biomolecules-10-00389-t001:** Galectin-3 expression in various diseases and potential use as clinical biomarkers.

	Diseases	Usage of Biomarker	Potential Use as Biomarkers	Refs.
Heart disease	acute heart failure	plasma level	combination with natriuretic peptide	[62]
	acute heart failure	plasma level	promising prognostic marker	[63]
	chronic heart failure	plasma level	useful in HF patients with preserved LVEF	[64]
	chronic heart failure	myocardial and plasma level	not associated with histology	[65]
	acute myocardial infarction	serum level	no definite relationship with ventricular remodeling	[66]
Nervous system diseases	myelin degeneration	tissues	activation of the phagocytosis of degenerated myelin	[42]
	intracerebral hemorrhage	plasma level	prognostic predictor	[67]
	subarachnoid hemorrhage	plasma level	prognostic predictor	[68]
	global brain ischemia	cerebrospinal fluid	prognostic marker and inflammatory mediator	[69]
Renal disease	CKD	plasma level	associated with progression of CKD	[25]
	SLE nephritis	serum and biopsy specimens	associated with SLE patients, particularly in SLE nephritis	[70]
Liver disease	liver fibrosis	serum Gal-3 related binding protein	assessing liver fibrosis	[71]
Connective tissue diseases	rheumatoid arthritis	serum level	increased in early rheumatoid arthritis	[72]
	SLE	serum anti-Gal-3 antibody	a key role in SLE skin lesions	[73]
	systemic sclerosis	serum level	predictor of mortality	[74]
Neoplasms	colorectal cancer	serum and tissues	related to tumor progression	[52]
	breast cancer	human cell lines	important factor for treatment	[75]
	non-small cell lung cancer	tumor expression	promotion of invasion and metastasis	[76]
	lung and prostate cancers	tumor expression	therapeutic target of tumor immunity	[77]
	cervical cancer	tumor expression	targets of multifunctional cancer treatment	[78]
	thyroid cancer	tumor expression	diagnostic marker	[79]

**Table 2 biomolecules-10-00389-t002:** Galectin-3 expression and findings as biomarkers in animal models.

	Animal Models	Experimental Methods	Experimental Findings	Refs.
Heart disease	chronic heart failure	intrapericardial injection of recombinant Gal-3	myocardial fibrosis and its pharmacological inhibition	[80]
	chronic heart failure	intrapericardial infusion of low-dose Gal-3	increased Gal-3 in hypertrophied hearts	[81]
	chronic heart failure	banding of the pulmonary artery	increase of Gal-3 in ventricles	[82]
	acute heart failure	viral myocarditis	time-course analysis of cardiac and serum Gal-3	[31]
Nervous system diseases	multiple sclerosis	EAE in Gal-3-deficient mice	reduction in severity of EAE	[83]
	multiple sclerosis	spinal cord in EAE	Gal-3 observed at early stage before symptoms	[84]
	Huntington’s disease	mouse model of Huntington’s disease	plasma and brain Gal3 levels correlated with disease severity	[40]
	brain ischemia	time course analysis of Gal-3 in hippocampus	usefulness of early stage of Gal-3 for ischemic brain	[39]
	brain ischemia	temperature-dependent enhancement of Gal-3	hypothermic prevention of Gal-3	[30]
	cerebral infarction	cerebral artery occlusion	up-regulated Gal-3 in late stage of permanent ischemia	[85]
Renal disease	renal ischemia	Gal-3 deficient mice	less acute renal tubular necrosis	[86]
	renal fibrosis	unilateral ureteral obstruction	renal fibroblast activation by macrophage-secreted Gal-3	[87]
	renal ischemia	unilateral ureteral obstruction	Gal-3 protect renal tubules	[88]
	lipid-induced renal injury	Gal-3 deficient mice	more marked in Gal-3 deficient mice	[89]
Liver disease	NAFLD/NASH	Gal-3 deficient mice	attenuated inflammation and fibrosis	[90]
Connective tissue diseases	SLE	anti-Gal-3 antibody injected into skin	induction of lupus-like histologic changes	[73]
Neoplasms	glioma	analysis of preneoplastic lesions	expressed in preneoplastic lesions	[43]
	ovarian cancer	ovarian cancer xenografted mice.	galectin-3 maintains ovarian cancer stem cells	[91]
	oral cancer	Gal-3 deficient mice	no significant difference	[92]

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
