# Peer review of "Galectin-3 as a Next-Generation Biomarker for Detecting Early Stage of Various Diseases"

_biomolecules, 2020, doi:10.3390/biom10030389_

Round 1
Reviewer 1 Report
The review paper describes the multifunctionality of the chimera type lectin galectin-3. Although, the manuscript is easy to follow and well organized, I am missing valuable and important data that have to be included prior the paper may be recommended for puplication.
Major points
The paper focuses on describing Gal-3 levels, association with inflammation, fibrosis, etc. I agree with that, it is the aim of the study. However, brief information regarding the mechanisms and signaling of Gal-3 in mentioned diseases and pathological conditions would improve the quality of the paper and provide to the readers a better overview of its function. Also please include a short chapter regarding its function in skin (the largest organ) diseases and repair.Minor point
Figure 2 – the authors listed in this figure are not listed at the end of the manuscript within the Reference list. It should be corrected.Author Response
Also please include a short chapter regarding its function in skin (the largest organ) diseases and repair.
Reply:
Thank you very much for the Reviewer’s valuable comment.
The short chapter "6.4. Skin diseases and its repair" has been added as the last paragraph of the section "6. Gal-3 in other diseases". (Lines 450-467)
Figure 2 – the authors listed in this figure are not listed at the end of the manuscript within the Reference list. It should be corrected.
Reply:
The indicated references in Figure 2 has been corrected and listed in "Reference section" at the end of the manuscript according to the Reviewer’s advice.

Reviewer 2 Report
The review summarizes published data on galectin-3 perspectives to be used as biomarker in clinical diagnostics and animal models. Currently, it is of great interest to study new biological markers that can be useful for monitoring the effectiveness of pharmacotherapy (personalized medicine), early diagnosis of diseases, prognoses of its clinical outcomes and play an important role in stratifying patients' risk. In this aspect, galectin-3, as a protein that is expressed by many cells, including neutrophils, macrophages, mast cells, fibroblasts and osteoclasts and found in the lungs, stomach, intestines, uterus and ovaries, is of significant interest for improving methods for diagnosis of diseases. The review is written in a good, accessible language and its originality is more than 85%.
I recommend the acceptance of the manuscript after minor considerations.
I would like to suggest authors to add more information about methods of detection of galectin-3. I think it will be benefit for manuscript. It will be also interesting to know about commercial test systems and therapeutics available in clinics. More figures presenting the differences in galectin-3 level between disease and normal will be also benefit visualize the data. Line 131-132: “On the other hands” should be replaced with “On the other hand”. Please check English. Line 214: degree designation is “oC”, but not “◦C”.Author Response
I would like to suggest authors to add more information about methods of detection of galectin-3. I think it will be benefit for manuscript. It will be also interesting to know about commercial test systems and therapeutics available in clinics.
Reply:
According to the Reviewer’s suggestions, the following sentences about the methods of detection of galectin-3 have been added at the top of the last paragraph of the section "2. Clinical significance and applications of galectine-3 (Gal-3)". (Lines 106-110)
Serum levels of Gal-3 are usually determined by enzyme-linked immunosorbent assay (ELISA) using commercially available materials [57,58]. The localization or time-course of Gal-3 expression in various organs are examined immunohistochemically using commercially available anti-Gal-3 antibodies [31].
More figures presenting the differences in galectin-3 level between disease and normal will be also benefit visualize the data.
Reply:
The papers indicating the galectin-3 level between disease and normal have been cited as references #57 and #58. Disease and normal serum levels of Gal-3 are usually determined by enzyme-linked immunosorbent assay (ELISA) using commercially available materials. And reference #31 shows the time-course analysis of serum galectin-3 in animal myocarditis including normal condition.
We appreciate the Reviewer #2’s evaluation of our manuscript.

Reviewer 3 Report
I really liked this review article. I feel that it is well written and highly appropriate for publication in Biomolecules. I expect that this paper will be heavily cited because it ties together many references to really look at the utility of galectin-3 as a biomarker of inflammatory diseases.
My one suggestion: the authors correctly note that galectin-3 has the strong possibility of being readily detectable at an early stage of many diseases in serum or in plasma. However, it would be helpful if the authors gave their perspectives regarding the disadvantage that comes with the fact that galectin-3 seems to be a global marker of inflammation. How could galectin-3 be used as a biomarker for heart disease, for example, when it is also upregulated with renal disease (or any number of other diseases)? Won't the broad spectrum upregulation of galectin-3 during inflammatory responses preclude its use in early detection of specific diseases?
Author Response
However, it would be helpful if the authors gave their perspectives regarding the disadvantage that comes with the fact that galectin-3 seems to be a global marker of inflammation.
Reply:
There are some disadvantages that come with the fact that galectin-3 seems to be a global marker of inflammation, as mentioned in reviewer #3's comments.
As described at Lines 208-211 in the manuscript:
In clinical setting, since various stages and degrees of myocardial inflammation and fibrosis may be present in a patient with heart disease, serum Gal-3 levels may reflect a sum of different pathophysiological conditions.
As described at Lines 211-213 in the manuscript:
Thus, it may be that the circulating Gal-3 levels in a patient with various stages of heart disease will not adequately reflect myocardial inflammation and fibrosis.
As described at Lines 567-569 in the manuscript:
Since Gal-3 levels are altered by different clinical factors depending on the underlying pathological conditions in patients, Gal-3 itself is not a disease-specific marker.
How could galectin-3 be used as a biomarker for heart disease, for example, when it is also upregulated with renal disease (or any number of other diseases)?
Reply:
Gal-3 expression is detected in various disease conditions such as heart disease, kidney disease, diabetes mellitus, viral infection, autoimmune disease, neurodegenerative disorders and tumor formation. This means that Gal-3 itself is not a disease-specific marker. However, Gal-3 is a potential biomarker for targeting early stage at a single pathological condition of several diseases.
As described at Lines 365-367 in the manuscript:
The wide tissue distribution of Gal-3 expression associated with fibrosis in both heart disease and kidney disease complicates the utility of Gal-3 as a cardiac biomarker in chronic kidney disease (CKD) patients.
However, as described at Lines 370-372 in the manuscript:
Higher concentrations of Gal-3 may be associated with progression of CKD, indicating potential novel mechanisms related to Gal-3 expression that may contribute to the progression of CKD.
Won't the broad spectrum upregulation of galectin-3 during inflammatory responses preclude its use in early detection of specific diseases?
Reply:
As mentioned above, Gal-3 itself is not a disease-specific marker. However, Gal-3 is a potential biomarker for targeting early stage at a single pathological condition of several diseases. Thus Gal-3 is not useful in complicated disease conditions including multiple inflammatory responses, as mentioned in reviewer #3's comments.
We appreciate the Reviewer #3’s comments and recommendations to our manuscript.
